# Emissive Pentacene-Loaded βcyclodextrin-Derived C-Nanodots Exhibit Red-Light Triggered Photothermal Effect

**DOI:** 10.3390/pharmaceutics17050543

**Published:** 2025-04-22

**Authors:** Ludovica Maugeri, Giorgia Fangano, Ester Butera, Giuseppe Forte, Paolo Giuseppe Bonacci, Nicolò Musso, Francesco Ruffino, Loredana Ferreri, Grazia Maria Letizia Consoli, Salvatore Petralia

**Affiliations:** 1Department of Drug and Health Sciences, University of Catania, Via Santa Sofia 64, 95125 Catania, Italy; ludovica.maugeri@phd.unict.it (L.M.); giorgia.charly@gmail.com (G.F.); ester.butera@phd.unict.it (E.B.); gforte@unict.it (G.F.); 2Department of Biomedicals and Biotechnologies Sciences, University of Catania, Via Santa Sofia 89, 95123 Catania, Italy; paolo.bonacci@phd.unict.it; 3Faculty of Medicine and Surgery, “Kore” University of Enna, Contrada Santa Panasia, 94100 Enna, Italy; nicolo.musso@unikore.it; 4Department of Physic and Astronomy, University of Catania, Via Santa Sofia 64, 95125 Catania, Italy; francesco.ruffino@dfa.unict.it; 5CNR-Institute of Biomolecular Chemistry, Via Paolo Gaifami 18, 95126 Catania, Italy; loryferreri@yahoo.it (L.F.); grazia.consoli@icb.cnr.it (G.M.L.C.); 6NANOMED, Research Centre for Nanomedicine and Pharmaceutical Nanotechnology, University of Catania, Viale A. Doria 6, 95124 Catania, Italy; 7CIB-Interuniversity Consortium for Biotechnologies U.O. of Catania, Via Flavia, 23/1, 34148 Trieste, Italy

**Keywords:** carbon nanodots, photothermal effect, photoluminescence, pentacene, cyclodextrin

## Abstract

**Background:** The design of multifunctional carbon based nanosystems exhibiting light-triggered hyperthermia, emission, low cytotoxicity, and drug delivery capability is of significant interest in the area of nanomaterials. In this study, we present red-emitting and photothermal carbon nanodots (Cdots-βCD/PTC) obtained by the encapsulation of hydrophobic pentacene (PTC) within Carbon nanodots (Cdots) synthesized from beta-cyclodextrin (βCD). **Methods:** The prepared nanostructures were investigated in terms of morphology, size, and optical properties, by absorption and emission optical spectroscopy, atomic force microscopy, dynamics light scattering, Z-potential, nuclear magnetic resonance, and infra-red spectroscopy. Molecular modelling simulation was used to investigate the geometry and the stabilization energy of the Cdots-βCD/PTC inclusion complex. **Results:** The as prepared Cdots-βCD/PTC demonstrated good water dispersibility, green-emission (ϕ_PL_ = 1.7%), and photothermal conversion (η = 17.4%) upon red-light excitation (680 nm). Furthermore, Cdots-βCD/PTC low cytotoxicity in the range 0.008 μg–0.8 μg and good interaction with albumin protein (K_SV_ = 2.78 ± 0.28 mL mg^−1^) were demonstrated. Molecular simulation analysis revealed the formation of the inclusion complex with an energy of −5.32 kcal mol^−1^, where PTC is orthogonally oriented in the βCD cavity. **Conclusions:** The results presented in this work highlight the potential of Cdots-βCD/PTC as a novel versatile nanosystem for biomedical applications, such as bioimaging and site-specific photothermal treatment of cancer cells.

## 1. Introduction

Carbon-based nanostructured materials, such as nanodots, quantum dots, nanoparticles, and nanosheets, have emerged as an attracting class of materials due to their unique physicochemical properties, remarkable optical behavior, efficient photothermal conversion under visible light, low toxicity, and promising biomedical applications [1]. Although carbon nanodots (Cdots) have been widely explored for photothermal and photodynamic therapies, only a limited number of Cdots-based nanohybrids have been designed to simultaneously integrate photothermal, photodynamic, luminescent, and nanocarrier functionalities [2].

Cdots are nanomaterials composed of a graphitic carbon core covered by an outer shell; the shell composition depends on the precursor and the preparation method used. These materials are particularly attractive due to their optical and photothermal properties, along with their low toxicity, high permeability, rapid clearance from the body, and ability to evade the immune system, making them well-suited for biomedical applications. Examples reported in the literature include applications for drug delivery [3], bioimaging [4], and photothermal therapy (PTT) [5].

PTT uses photothermal agents to generate heat via photon energy absorption, which leads to the thermal ablation of target cells and subsequent cell death. Effective photothermal agents should exhibit strong light absorption and effective photoconversion properties [6]. To enhance the efficacy of PTT, carbon-based nanomaterials are often combined with photothermal plasmonic nanoparticles to form hybrid nanomaterials such as graphene-oxide/gold and silver nanoparticles; however, their high cost and complex synthesis processes pose significant limitations [7]. Therefore, alternative low-cost photothermal agents are being explored.

Owing to their distinctive electrical, optical, and photophysical characteristics, pentacene (PTC) and its derivatives are very attractive compounds that have been extensively studied in materials science. These molecules exhibit excellent photothermal properties, strong emissive behavior, and effective photosensitizing capabilities [6]. Furthermore, PTC can interact with oxygen in a steady state, exhibiting both energy and electron transfer properties. However, its poor water solubility and tendency to aggregate present significant challenges, limiting its application as a photo-responsive agent in biomedical fields. A well-established strategy to overcome these limitations involves the encapsulation of hydrophobic molecules within nanocarriers, thereby enhancing their aqueous dispersibility and bioavailability [8].

Despite their potential for multifunctionality, cyclodextrins (CDs) have been relatively underexplored. β-cyclodextrin (βCD) consists of seven glucopyranose units linked by α-1,4-glycosidic bonds, resulting in a characteristic truncated cone-shaped structure. The cyclic structures of CDs enable them to encapsulate a wide range of guest molecules through the formation of host-guest inclusion complexes. CDs have been widely recognized as safe solubilizing agents, particularly for Biopharmaceutics Classification System (BCS) class II and IV drugs, and are already incorporated into several approved pharmaceutical formulations [9].

Recently, our research group developed biocompatible, water-dispersible, luminescent carbon nanodots (Cdots-PNMs) through a simple thermal treatment of the thermo-responsive poly(*N*-isopropylacrylamide) polymer (PNM), which effectively serve as nanocarriers for PTC [6]. These nanohybrid systems exhibit strong fluorescence emission (enabled by surface passivation of Cdots) and photo- and thermo-responsive behavior, inducing effects such as photothermia, photocatalysis, and controlled drug release under light stimulation. Furthermore, they can be synthesized through a green process without the use of organic solvents, reagents, or additives, and demonstrate photosensing capabilities triggered by biocompatible red-light [10]. In this study, we focus on a novel photothermal, luminescent, and water-dispersible carbon nanodot system derived from βCD (Cdots-βCD). The prepared nanosystem, composed of an emissive C-sp^2^ core and an outer βCD shell, offers the advantage of combining the luminescent and photothermal properties of carbon dots with the host properties of βCD for drug delivery.

Our experimental and simulation investigations confirm the effective encapsulation of PTC to obtain the Cdots-βCD/PTC nanosystem. This nanosystem is prepared as depicted in Figure 1 and fully characterized using different techniques, including optical absorption and emission spectroscopy, Fourier transform infrared (FTIR) and nuclear magnetic resonance (NMR) spectroscopy, dynamic light scattering (DLS), Z-potential, and atomic force microscopy (AFM). The biocompatibility of the Cdots- βCD and Cdots- βCD/PTC nanosystems was detected by MTT assay on CaCo-2 cell line. The biological interaction with Bovine Serum Albumin (BSA) was also studied by steady-state fluorescence emission.

## 2. Materials and Methods

All chemicals were obtained from Sigma Aldrich at the highest possible purity and were used as received. Milli-Q-grade water was used in all preparations. All solvents were spectrophotometric grade.

### 2.1. Preparation of Cdots-βCD

Beta-cyclodextrin (βCD) (200 mg) was thermally treated at 190 °C in air for 3 h. The resulting dark-reddish solid was washed with 1 mL of Milli-Q water, then suspended in water and sonicated for 5 min. To effectively remove the precursor excess and byproducts, the water dispersion was centrifugated at 13,000 rpm for 5 min and the supernatant was filtered through a 0.2 μm pore membrane filter. The resulting red-yellowish colloidal solution containing Cdots-βCD was further purified through dialysis in Milli-Q water using a dialysis membrane with a 10 kDa molecular weight cutoff for 24 h.

### 2.2. Preparation of Cdots-βCD/PTC Nanohybrid

An excess of pentacene (PTC) (10 mg) was added to an aqueous solution of Cdots-βCD (5 mg mL^−1^), and the mixture was stirred at room temperature for 3 days in the dark. The excess solid pentacene was removed by centrifugation (13,000 rpm for 10 min). The resulting supernatant, containing Cdots-βCD/PTC, was separated using a pellet of unencapsulated PTC and subsequently filtered through a 0.2 μm pore membrane.

### 2.3. Instrumentation

#### 2.3.1. UV-Vis Spectroscopy

UV-Vis optical absorption spectra were produced to assess both the formation of Cdots-βCD and Cdots-βCD/PTC. The spectra were recorded on a Perkin-Elmer 365 spectrophotometer (Long Island, NY, USA) using quartz cuvettes with a path length of 10 mm.

#### 2.3.2. Fluorescence Spectroscopy

Fluorescence emission spectra were performed to verify whether Cdots-βCD exhibited the typical excitation-dependent behavior of carbon-nanodots and to confirm the formation of the Cdots-βCD/PTC complex. The measurements were conducted with different excitation wavelengths: from 360 to 520 nm for Cdots-βCD fluorescence emission (3 nm/3 nm slit width) and from 640 to 680 nm for PTC fluorescence emission (5 nm/5 nm slit width). The fluorescence spectra were acquired on a Horiba spectrofluorometer using a quartz cuvette with an optical length of 10 mm. Steady-state fluorescence spectra of bovine serum albumin (BSA) with different concentrations of Cdots-βCD were recorded on a Horiba Spectra-Max spectrofluorophotometer (Horiba, Rome, Italy). The excitation wavelength was set at 280 nm and the emission spectra was recorded in the wavelength range of 300−450 nm (slits 2/2). A quartz cuvette with an optical length of 10 mm was used. The amount of protein concentration was 4 mg mL^−1^, and the Cdots-βCD concentrations for titration were 0.0, 0.048, 0.094, 0.14, 0.186, 0.23, 0.318, 0.403, 0.485, 0.565, and 0.642 mg mL^−1^.

#### 2.3.3. ATR-FTIR Spectroscopy

The effective formation of Cdots-βCD and Cdots-βCD/PTC was also investigated by ATR-FTIR spectroscopy (Perkin-Elmer, Long Island, NY, USA). A Perkin-Elmer Spectrum Two Fourier Transform Infrared spectrophotometer with both transmission and attenuated total reflectance modules was used.

#### 2.3.4. Atomic Force Microscopy (AFM)

AFM analyses were performed using a Bruker-Innova microscope operating in high amplitude mode and ultra sharpened Si tips (MSNL-10 from Bruker Instruments (Karlsruhe, Germany), with anisotropic geometry, radius of curvature 2 nm, tip height 2.5 nm, front angle 15°, back angle 25°, side angle 22.5°, resonance frequencies of 90–160 kHz, nominal spring constant of 0.07 N/m) were used and substituted as soon as a loose resolution was observed during the acquisition. The images were reconstructed by acquiring 512 × 512 lines, scan rate 0.8 Hz. The AFM images were analyzed by using the SPMLABANALYSES V7.00 software. Samples were prepared by drop casting onto pre-cleaned silicon substrate and were allowed to dry under gentle nitrogen blow-drying to ensure proper adsorption.

NMR (Nuclear Magnetic Resonance) spectra were acquired on a Bruker 400^TM^ spectrometer (Bruker, Germany) and calibrated on the solvent residual proton (HOD, 4.72 ppm).

#### 2.3.5. DLS and Zeta Potential Measurements

DLS and Z-potential analyses confirmed the formation of nanosized and slightly negatively charged Cdots-βCD structures. The measurements were performed by a Zetasizer NanoZS90 analyzer (Malvern Instrument, Malvern, UK) equipped with a 633 nm laser, at a scattering angle of 90° and 25 °C temperature. The size of the particles was calculated from the diffusion coefficient using the Stokes−Einstein equation. The DLS distributions are reported in Appendix A.

### 2.4. Photothermal Properties

The photothermal properties of Cdots-βCD and Cdots-βCD/PTC were investigated by irradiating glass tubes (3 mm in diameter) containing varying amounts of nanostructure dispersions. Specifically, 100 µL of the Cdots-βCD dispersion was exposed to a 405 nm continuous-wave (CW) laser at different power levels (50, 100, 150, and 212 mW) for 10 min, and the temperature was monitored using a FLIR infrared thermal imaging camera (FLIR, Limbiate, Italy). In a similar setup, a 100 µL volume of the Cdots-βCD/PTC dispersion was irradiated with a 680 nm CW laser at 810 mW power for varying durations. The temperature of the solution was measured every 10 s during both the heating and cooling processes using the FLIR infrared thermal imaging camera. For each experiment, three replicates were performed.

### 2.5. Modeling Simulation

Three geometries of pentacene interacting with the β-cyclodextrin cavity were considered, as shown in Appendix A. Various initial configurations of these complexes underwent equilibration through 10,000 steps of minimization followed by 100 ns of Molecular Dynamics simulations under the PVT ensemble, using water as the solvent. During the final 10 ns, three structures were selected and further optimized (Appendix A). The Consistent Valence Force Field (CVFF) was applied, maintaining a temperature of 298 K and a pressure of 1 atm regulated by the Berendsen barostat [11]. The geometries obtained from molecular mechanics simulations were subsequently refined using Density Functional theory (DFT) with the B3LYP functional including D3 correction to account for non-covalent interactions, in combination with the triple zeta 6-311+G(d,p) basis set [12,13,14,15]. Solvent effects were incorporated using the polarizable Continuum Model while basis set superposition error (BSSE) in interaction energies was corrected using the counterpoise method. UV-vis spectra were simulated using CAM-B3LYP functional with the same basis set.

### 2.6. Cytotoxicity Tests

To evaluate the effect of Cdots-βCD and Cdots-βCD/PTC, MTT (3-(4,5-dimethylthiazol-2-yl)-2,5-diphenyltetrazolium bromide) assays were performed as previously described. Briefly, human colorectal adenocarcinoma cells (CaCo-2 HTB-37TM, American Type Culture Collection, Manassas, VA, USA) were grown in Dulbecco’s MEM (DMEM) with 10% heat-inactivated fetal bovine serum, 2 mM L-Alanyl-L-Glutamine, and penicillin-streptomycin (50 units-50 μg for mL) and incubated at 37 °C in a humidified atmosphere of 5% CO_2_, 95% air. CaCo-2 cells were plated in 96 well pates at concentration of 5000 cells/well and incubated at 37 °C for 24 h before treatment. For Cdots-βCD/PTC, 0.8 mg were weighed and diluted in 100 μL of PBS 1×, thus obtaining a stock solution of 8 μg/μL. Three serial dilutions were then created: 80 ng/ μL, 8 ng/ µL, and 0.8 ng/ µL. Cells were treated with 10 µL of each solution. Untreated cells were used as controls. Microplates were incubated at 37 °C in a humidified atmosphere of 5% CO_2_, 95% air for 24 h. Cytotoxicity was measured with colorimetric assay based on the use of tetrazolium salt MTT (3-(4,5-dimethylthiazol-2-yl)-2,5-diphenyl tetrazolium bromide). The results were read on a multiwell scanning spectrophotometer (BioTek Synergy H1 Multimode Reader, BioTek, Winooski, VT, USA), using a wavelength of 569 nm. Each value was an average of four wells. To evaluate the impact of Cdots-βCD/PTC complex on cellular metabolism, the percentage of growth compared to the control was calculated, using the difference in absorbance by the wells treated with the dispersion. All statistical analyses were performed using GraphPad Prism 6.0 software.

## 3. Results and Discussion

### 3.1. Cdots-βCD Preparation and Characterization

Cdots-βCD nanostructures were prepared using a one-pot mild-thermal process recently developed by our team, using βCD as the unique precursor without any chemical reagents or solvents. [10] The UV-Vis optical absorption spectra of Cdots-βCD revealed characteristic absorption bands typical of aromatic carbon nanodots. Specifically, a well-defined band centered at around 270 nm was observed, corresponding to the π−π* transition originating from sp^2^-hybridized carbon structures (Figure 1A) [3,10]. An increase in the optical absorption intensity of the π−π* transition was detected at different reaction times (45 min, 1.5 h, and 3.0 h), indicating a progressive growth in nanostructure size as the reaction proceeded (Appendix A). The optical band gap was calculated by Tauc’s plot, which showed the variation of (Ass hυ)^1/2^ vs. (hυ) for Cdots-βCD (inset Figure 1A). The optical energy band gap for the allowed direct transitions was estimated to be about Eg = 3.49 eV. This is in good agreement with carbonaceous nanodots reported in the literature [16].

Figure 1B presents the fluorescence emission spectra of Cdots-βCD recorded at excitation wavelengths ranging from 360 to 520 nm. The Cdots-βCD nanostructures exhibited excitation-dependent emission behavior, a characteristic of carbon-based nanodots. The photoluminescence quantum yield (ϕPL = 1.7%) of the Cdots-βCD dispersion in water (refractive index, n = 1.33) was determined using quinine sulfate as a standard reference (n = 1.36, ϕPL = 0.56) with an excitation wavelength of 360 nm. ATR-FTIR spectroscopy of Cdots-βCD (Figure 1C, red line) revealed the same characteristic diagnostic peaks of the βCD precursor (Figure 1C, black line) according to the literature data for other cyclodextrin-derived carbon nanodots [6]. Signals at around 3289 cm^−1^ and 1648 cm^−1^ could be related to the O-H stretching and bending, respectively, for both Cdots-βCD and βCD. A signal around 2919 cm^−1^ is diagnostic for the C-H stretching of the βCD cavity on nanostructures and precursor; signals at 1148 cm^−1^ and 1018 cm^−1^ can be related to the C-O-C vibration band and C-O stretching, respectively.

Morphological analysis of Cdots-βCD by AFM, in Figure 1D, revealed the presence of an isolated spherical carbon core (height of 2.5 ± 0.2 nm) and larger aggregates. Further AFM images and section profile analyses are reported in Appendix A.

DLS measurements of the Cdots-βCD dispersion in water revealed a dominant population of nanostructures with a mean hydrodynamic diameter of 7.12 ± 3.6 nm (100% in volume distribution mode) and a Z-potential of −9.80 ± 0.984 mV. In contrast, the water dispersion of native β-CD exhibited smaller structures with a mean hydrodynamic diameter of 1.78 ± 0.44 nm (100% in volume distribution mode) and a Z-potential of −1.87 ± 1.76 mV. The presence of an intact βCD shell covering the carbon core was confirmed by the ^1^H NMR spectrum, which displayed signals in the characteristic regions of the βCD precursor, including the diagnostic H3 and H5 proton peaks located within the cyclodextrin cavity at 3.86 and 3.71 ppm (Figure 1E).

To assess the interaction between Cdots-βCD and bovine serum albumin (BSA), steady-state fluorescence measurements were performed in phosphate buffer. The fluorescence intensity of BSA (4 mg mL^−1^) was quenched upon adding different concentrations of Cdots-βCD (0.0, 0.048, 0.094, 0.14, 0.186, 0.23, 0.318, 0.403, 0.485, 0.565, and 0.642 mg mL^−1^) without a shift of the maximum emission wavelength at 350 nm (Figure 2A). The observed decrease in intensity suggests that Cdots-βCD can interact with BSA and the lack of a shift in emission wavelength implies that the carbon dots have minimal impact on the microenvironment of the tryptophan residue [17,18]. Fluorescence quenching followed the well-known Stern-Volmer equation expressed as:$$\frac{F_{0}}{F}=1+kq\;t_{0}\left[Q\right]=1+K_{SV}[Q]$$
where F_0_ and F are the fluorescence in the absence and presence of the quencher (Cdots-βCD), respectively, [Q] is the Cdots-βCD concentration, and K_SV_ is the Stern–Volmer constant, which indicates the quenching efficiency. Figure 2B shows the quenching behavior from 0 to 0.642 mg ml^−1^ concentration aligned with the Stern–Volmer equation [19]. The result showed a K_SV_ value of 2.78 ± 0.28 mL mg ^−1^ at 25 °C.

To investigate the photothermal properties of Cdots-βCD, a 100 μL aliquot of water Cdots-βCD dispersion (0.5 mg mL^−1^) in a glass tube was exposed to 405 nm laser irradiation with different laser powers (50, 100, 150, and 215 mW). The laser power-dependent photothermal behavior was confirmed as shown in Figure 3, where the recorded temperature rise values were around 2, 2.9, 4.2, and 6.8 °C, respectively. The system exhibits reversible heating and cooling cycles. During each heating step, the temperature increases about 6 °C for A_405nm_ = 0.56, before gradually decreasing to 23 °C, during the cooling step. Photothermal experiments for the reference (βCD precursor) have been reported (Figure 3A green line). Using the Roper’s equation (Appendix A), a photothermal conversion efficiency (η) of about 17.4% and a time constant (τ_s_) of 130.02 s were calculated.

### 3.2. Cdots-βCD/PTC Preparation and Characterization

The preparation of the Cdots-βCD/PTC inclusion complex was performed by a simple solubility phase method. In brief, an excess of PTC (10 mg) was stirred in an aqueous solution of Cdots-βCD (5 mg mL^−1^) at room temperature for 3 days in the dark. The solid unentrapped pentacene was removed by centrifugation (13,000 rpm for 10 min). Specifically, the supernatant containing Cdots-βCD/PTC was separated by the pellet (containing unentrapped PTC) and subsequently filtered through a 0.2 μm pore membrane.

The UV-Vis optical absorption and emission spectra of the Cdots-βCD/PTC system are shown in Figure 4A. The UV-Vis spectrum revealed a broad absorption feature with a prominent peak around 272 nm and a broad peak at around 345 nm related to the ππ* nπ* transition of the carbon core. Additional absorption bands in the range from 550 to 1000 nm indicate the presence of PTC entrapped into the Cdots-βCD nanostructures. The absorption bands across the visible and NIR regions suggest potential applications in bioimaging and PTT, as these regions are relevant for biological transparency.

The emission spectra show a strong green luminescence signal at around 500 nm (upon excitation wavelength 400 nm) generated by the C-core in Cdots-βCD/PCT. This emission is enhanced and red-shifted (by about 25 nm) compared to the emission observed for Cdots-βCD (Appendix A), indicating an effective interaction between Cdots-βCD and PTC. Additional emission bands at around 800–820 nm related to the PTC excitation (680 nm) were observed to again confirm the presence of the PTC in Cdots-βCD/PCT nanostructures. ATR-FTIR spectroscopy of Cdots-βCD/PTC was performed to analyze the chemical composition and interaction within the system. The PTC spectrum, in blue in Figure 4B, shows characteristic absorption bands, including sharp peaks below 1000 cm^−1^, which may correspond to specific stretching vibrations of the PTC functional groups. The Cdots-βCD FTIR spectrum, in red, shows specific peaks of the βCD functional groups as described in the previous section (Figure 1C).

The FTIR spectrum of the Cdots-βCD/PTC hybrid system (Figure 4B, black line) shows clear shifts in the absorption bands within the 1500–1700 cm^−1^ range, suggesting interactions between PTC and the cyclodextrin cavity. Specifically, the Cdots-βCD/PTC spectrum displays characteristic signals attributable to both Cdots-βCD and PTC. The following bands confirm the presence of Cdots-βCD nanostructures: 3289 cm^−1^ (O–H stretching), 1648 cm^−1^ (C=O stretching), 2919 cm^−1^ (C–H stretching), 1465 cm^−1^ (C–H bending), 1148 cm^−1^ (C–OH stretching), and 1018 cm^−1^ (C–O stretching). In contrast, the shifted peaks at 3042 cm^−1^ (C–H of PTC), 1550 cm^−1^ (aromatic C=C stretching), 907–850 cm^−1^ (aromatic C–H stretching), and 719–463 cm^−1^ (aromatic C=C–C bending) indicate the presence of PTC. For comparison, the FTIR spectrum of the PTC precursor is shown in Figure 4B (blue line), displaying its characteristic peaks: 3070 cm^−1^ (C–H stretching), 1670–1350 cm^−1^ (conjugated C=C stretching), 1220–960 cm^−1^ (C–H vibrations), 892–839 cm^−1^ (aromatic C–H stretching), and 719–463 cm^−1^ (aromatic C=C–C bending) [20]. Likewise, the Cdots-βCD spectrum (Figure 4B, red line) aligns with previously reported FTIR data for cyclodextrin-derived carbon nanodots [3].

DLS investigation of the Cdots-βCD/PTC dispersion in water revealed the presence of large aggregates (388.43 ± 36.86 nm, (100% Volume) probably due to the hydrophobic interactions between PTC species in water dispersion. A Z-potential value of −38.80 ± 3.03 mV was obtained for Cdots-βCD/PTC dispersion in water. Morphological analysis of the Cdots-βCD/PTC nanosystem was performed by AFM. Figure 4C shows the presence of spherical carbon cores (8.2 ± 0.8 nm), and larger aggregates have also been detected, confirming DLS data. Additional AFM images and section profile analysis are reported in Appendix A.

The photothermal behavior of the Cdots-βCD/PTC complex was investigated and confirmed by irradiating a 100 μL aliquot of Cdots-βCD/PTC (A_680nm_ = 0.0815) with a CW laser (excitation wavelength 680 nm and power value 810 mW). The temperature increase recorded by a thermocamera after 5 min of irradiation was about 6.3 °C (Figure 5A). The system exhibited a reversible photothermal process as shown in Figure 5B. In detail, during the heating phase, the temperature rapidly increased from room temperature (T_env_ = 23 °C) to the maximum temperature (T_max_) of about 29.9 °C, then the laser was turned off and the temperature dropped back to room temperature (Figure 5B). The observed photothermal cycles highlighted the capability of the Cdots-βCD/PTC complex to effectively absorb and dissipate thermal energy under these experimental conditions. A photothermal conversion efficiency (η) of about 14.0% and a time constant (τ_s_) of 148.34 s were calculated using the Roper’s equation (Appendix A). Representative thermographs of the nanostructure dispersions during the photothermal experiments in the glass tube holder are shown in Figure 5A,B.

To assess the stability of pentacene complexes with the β-cyclodextrin cavity, the free energy values for complex formation were calculated using the following equation:$${∆G}_{f}=(G_{cyclodestrin-pentacene})-(G_{cyclodextrin\;}+G_{pentacene})$$

The optimized structures of the complexes obtained by DFT calculations are depicted in Appendix A, while the calculated Δ*G_f_* values (in kcal mol^−1^) are reported in Appendix A. These results clearly indicate that the most thermodynamically stable complexes, based on the Δ*G_f_* values, correspond to species vertically intercalated in the cyclodextrin cavity. Specifically, complex C1 exhibits a Δ*G_f_* of −9.35 kcal mol^−1^, while C2 shows a similar free energy of formation value of −9.71 kcal mol^−1^. In contrast, the geometry in which pentacene lies parallel to the cyclodextrin cavity (C3) presents a less favorable Δ*G_f_* of −5.32 kcal mol^−1^. Notably, the C2 structure, in which the initial geometry pentacene is oriented orthogonally but largely external to the cyclodextrin cavity, undergoes significant structural evolution during molecular dynamics simulations. This transformation leads to the progressive internalization of pentacene, driven by its low affinity for the aqueous solvent. Among the possible geometries considered, the transition toward the C1 structure is particularly significant, since it represents the configuration that minimizes the surface exposure of pentacene to water.

The high stability of the Cdots-βCD/PCT complexes agreed with the calculated absorption spectrum (see Figure 6), whose data are reported in Appendix A.

### 3.3. Cytotoxicity Tests Cdots-βCD and Cdots-βCD/PTC

The optical properties of Cdots-βCD, in terms of luminescence and photothermal effect, encouraged us to investigate the safety of this nanocarrier. To this aim, we exposed Caco-2 cell line to increasing amounts of Cdots-βCD and Cdots-βCD/PTC (0.008 μg, 0.08 μg, and 0.8 μg) for 24 h. At the end of the treatment, the cell viability, measured by MTT assay, did not show any difference between untreated cells (control) and cells treated with Cdots-βCD. Similarly, cells treated with low concentrations of Cdots-βCD/PTC (0.08 μg and 0.008 μg) did not show a statistically significant reduction in cell growth compared to the control (*p* = 0.1876–0.7398), suggesting negligible cell mortality in this range (Figure 7A). Treatment with the highest concentration of Cdots-βCD/PTC (0.8 μg) resulted in a slight reduction in cell growth (about 20%) compared to the control (*p* < 0.0001) (Figure 7B). Statistical data were derived using GraphPad PRIMS 10 version 10.4.1 and the results are reported in Appendix A.

In summary, Cdots-βCD and Cdots-βCD/PTC nanostructures exhibited low cytotoxicity within the concentration range of 0.008–0.8 μg in Caco-2 cancer cells line.

## 4. Conclusions

In this work, we report the successful synthesis of a novel multifunctional nanosystem obtained by encapsulating pentacene (PTC) in carbon nanodots derived from simple heating of β-cyclodextrin (Cdots-βCD) is reported. The encapsulation approach allowed us to overcome the hydrophobicity and aggregation tendency of PTC, which limits its biological applications, while preserving its photophysical properties. Different techniques, including UV-Vis, fluorescence, ATR-FTIR and NMR spectroscopy, and AFM microscopy, have confirmed the formation of the Cdots-βCD/PCT nanohybrid system and its unique physicochemical properties, such as excellent water dispersibility, strong fluorescence emission, and efficient photothermal conversion. Molecular simulation investigation supported the formation of a host-guest inclusion complex with an energy of −5.32 kcal mol^−1^, where PTC is orthogonally oriented in the βCD cavity. Low cytotoxicity of Cdots-βCD/PTC in the range from 0.008 μg to 0.8 μg was confirmed by MTT assay on human cells. As a support for potential biological applications, the low cytotoxicity of the Cdots-βCD/PCT nanohybrid was demonstrated as well as its ability to interact with the albumin (Ksv = 2.78 ± 0.28 mL mg^−1^). Green and red fluorescence emission (around 500 nm and 800 nm) and photothermal conversion (η = 14.1%) triggered by biofriendly red-light irradiation suggest applications of Cdots-βCD/PTC in bioimaging and localized hyperthermia. The results presented in this work highlight the potential of Cdots-βCD/PTC as a novel versatile nanosystem for biomedical applications, such as visualization and site-specific photothermal treatment of cancer cells.

## Data Availability

All data are available on Appendix A.

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
