# Peer review of "Emissive Pentacene-Loaded βcyclodextrin-Derived C-Nanodots Exhibit Red-Light Triggered Photothermal Effect"

_pharmaceutics, 2025, doi:10.3390/pharmaceutics17050543_

Round 1
Reviewer 1 Report
Comments and Suggestions for Authors
The manuscript has several significant issues that need to be addressed before it can be considered for publication. The current state of the manuscript appears more like a draft than a polished submission, as the authors have not thoroughly checked the content for errors or adherence to the journal's guidelines. Below are the specific concerns:
-
There are too many typos in this article. The author is requested to read through the entire text carefully and check for typos, as they make the reading experience quite uncomfortable.
-
The font formatting (also including the references) is inconsistent. Please unify the font and formatting throughout the entire document.
-
The paragraph structure is disorganized, with some paragraphs consisting of just a single sentence!
-
The numerical annotations on the figures are also incorrectly formatted, such as errors like "17, 4%", and also many subscript, F0, which should be F0.
-
The abstract is poorly written. For instance, some instrument terms are abbreviated upon their first mention, whereas they should be written out in full the first time they appear.
-
There is an issue with the calculation of the P-value in Figure 7. For example, P<0.0001 indicates a highly significant difference between the values, but the figure shows "*" instead of "***," and this significant difference does not align with the description in the article. Please carefully re-check the calculation of the P-value and must provide a detailed method for how the P-value was calculated (which method or software).
Overall, this article is too rough and lacks careful refinement. Some technical issues should not appear in a submitted article. The author is requested to revise the article thoroughly.
Comments on the Quality of English LanguagePlease see the comments.
Author Response
The manuscript has several significant issues that need to be addressed before it can be considered for publication. The current state of the manuscript appears more like a draft than a polished submission, as the authors have not thoroughly checked the content for errors or adherence to the journal's guidelines. Below are the specific concerns:
- There are too many typos in this article. The author is requested to read through the entire text carefully and check for typos, as they make the reading experience quite uncomfortable.
Author’s response: thanks for your suggestion the manuscript was revised and the typos corrected
- The font formatting (also including the references) is inconsistent. Please unify the font and formatting throughout the entire document.
Author’s response: the font and size were corrected
2. The paragraph structure is disorganized, with some paragraphs consisting of just a single sentence!
Author’s response: thanks for your suggestion the manuscript was revised and just three main paragraphs have been reported
3. The numerical annotations on the figures are also incorrectly formatted, such as errors like "17, 4%", and also many subscript, F0, which should be F0.
Author’s response: Thanks for your comment we corrected the text and the figures as suggested
4. The abstract is poorly written. For instance, some instrument terms are abbreviated upon their first mention, whereas they should be written out in full the first time they appear.
Author’s response: Thanks for your comment we correct the text as suggested.
5. There is an issue with the calculation of the P-value in Figure 7. For example, P<0.0001 indicates a highly significant difference between the values, but the figure shows "*" instead of "***," and this significant difference does not align with the description in the article. Please carefully re-check the calculation of the P-value and must provide a detailed method for how the P-value was calculated (which method or software).
Author’s response: Thanks for your comment we revised the text, figure 7 and the table S2 and S3, to better explain the biological data.
Reviewer 2 Report
Comments and Suggestions for Authors
The manuscript shows the synthesis and characterization of beta-CD based nanodots. Furthermore the pentacene is added to the nanodots in order to enhance its solubility and use such host-guest complexes as photothermal agent.
The results are presented in a rather non-detailed manner, where discussion and similarities/differences obtained by certain methods are not discussed almost at all (for instance huge disparity between AFM and DLS results in case of pentacene nanodots). The methodology should be improved since control experiments for some methods were not done. From the start of the manuscript it is not clear why Cdots are used as a carrier and not some kind of untreated cyclodextrin. What does one gain by converting cyclodextrins into Cdots and is this thermal synthesis procedure such that some (minor/major) portion of beta-CD remains intact, which consequently enables host-guest interactions? The manuscript shows some interesting results, however, it should be only published after a major revision upon which the following issues should be addressed.
Lines 140-105: The Figure is not 2a, but 1a. Also, in the same sentence, all of a sudden there is a mention of HP-beta-CD?!
Lines 110-112: "The photoluminescence quantum yield (φPL =1.7%) of the Cdots- βCD dispersion in water (refractive index, n = 1.33) was determined using quinine sulfate as a standard reference (n = 1.36, φPL = 0.56)." Since spectra show a large dependence on the excitation wavelength, for which excitation wavelength was quantum yield calculated?
Lines 112-114: It would be beneficial to add a short discussion about the similarities and differences between FT-IR spectra of beta-CD and corresponding carbon dots.
It would also be useful to add DLS distributions to the supplementary information.
Lines 307-310: "The unentrapped pentacene was removed by centrifugation (13,000 rpm for 10 min) followed by a washing step at 3,000 rpm for 5 minutes. The final dispersion was subsequently filtered through a 0.2 μm pore membrane filter."
In this passage, it is not clear where unentrapped pentacene was? Was it in the pellet?
The materials and methods section lacks the whole part about albumin nanodots interaction, which should definitely be added. Especially, the excitation wavelength. It is not shown what kind of interaction do Cdots and BSA exhibit. The quenching of fluorescence can be both static and dynamic. The measurements should be repeated at three different temperatures to determine what kind of quenching this is. Maybe in this case it is a purely dynamic quenching. Potentially Cdots could possess unpaired electrons which could easily quench fluorescence. Also, authors should record spectra of just Cdots at the same excitation wavelength and in the same emission range to see if there would be some overlap of Cdots and BSA emission. Ksv values should have units of ml/mg.
Figure 3 - It would be nice to highlight the moment when the irradiation stops. To show that the effect arises mostly from Cdots, a control experiment should be done with water only or even beta-CD dissolved in water.
In Supplementary material "tau s" is mentioned in the text, but does not appear in the equation 4. Is tau or tau s supposed to be shown in all of the following graphs? In the main text, "pi s" denotes the time constant. The use of variable names is inconsistent. All the graphs in Figure S3 miss the letter theta in the x-axis title. In Supplementary material it is not explained what hA parameter stands for.
Lines 196-198 - "Additional emission bands at around 800-820 nm related to the PTC excitation (680 nm) were observed to again confirm the presence of the PTC in Cdots-βCD/PCT nanostrutcures" - A fluorescence spectrum at the 680 nm excitation should be shown for the Cdots without PTC as well.
Lines 211-213: "The PTC spectrum (blue curve) revealed these diagnostic peaks: 1700-1600 cm-1 (C=O stretching), 1570 cm-1 (aromatic C=O stretching), 1485 cm-1 (C=C stretching), 1350 cm-1 (C-N stretching), 1150-1100 cm-1 (C-Cl stretching)." Why would there be C-N and C-Cl stretching, if pentacene contains only carbons and hydrogens? The assignation should be redone and some more comments about FTIR spectra should be made.
Lines 214-219 - How is a 10-fold difference between DLS and AFM results possible, especially taking into account that the caption of Figure 4 states that the 3-D AFM image is a representative example.
Line 239 - Again time constant probably should not be "pi".
Line 253 - There is no Figure S11.
Why was the interaction between PTC Cdots and albumin not studied?
Conclusions - Lines 396-399 - "As a support for potential biological applications, the low cytotoxicity of the Cdots-βCD/PCT nanohybrid was demonstrated as well as its ability to bind albumin with a strong affinity (association constant of about 2.78 ± 0.28398
mg ml-1), which suggests a potential role of the nanohybrid in the formation of corone proteins." - The strong affinity for BSA binding in the view of previous comments was not proved and Ksv is not an affinity constant.
Lines 402-404 - "The results presented in this work, highlight the potential of Cdots-βCD/PTC as a novel versatile nanosystem for biomedical applications, such as visualization and site-specific photothermal treatment of cancer cells." - Since the colon cancer cell line was already used it would make sense to show using the same cell line the photothermal effect on cell viability.
Comments on the Quality of English LanguageThere are a lot of typos that need to be corrected after careful reading.
Section 3.3. Measurements are performed, not the spectra.
Line 330 -resolution loss, instead of "loose".
Conclusions - In this work, we report the successful synthesis of a novel multifunctional nanosystem obtained by encapsulating pentacene (PTC) in carbon nanodots derived from simple heating of β-cyclodextrin (Cdots-βCD) is reported.
This sentence is wrong with "we report" at the beginning and "is reported" at the end.
Author Response
1. Lines 140-105: The Figure is not 2a, but 1a. Also, in the same sentence, all of a sudden there is a mention of HP-beta-CD?!
Author’s response: thanks for your suggestion we have corrected the typos
2. Lines 110-112: "The photoluminescence quantum yield (φPL =1.7%) of the Cdots- βCD dispersion in water (refractive index, n = 1.33) was determined using quinine sulfate as a standard reference (n = 1.36, φPL = 0.56)." Since spectra show a large dependence on the excitation wavelength, for which excitation wavelength was quantum yield calculated?
Author’s response: thanks for your suggestion, in the revised manuscript we have included the excitation wavelength of 360nm used during the experiments.
3. Lines 112-114: It would be beneficial to add a short discussion about the similarities and differences between FT-IR spectra of beta-CD and corresponding carbon dots.
Author’s response: thanks for your suggestion we have modified the manuscript to better explain the FTIR analysis.
4. It would also be useful to add DLS distributions to the supplementary information.
Author’s response: The DLS distributions have been reported in SI (figure S9 and S10).
5. Lines 307-310: "The unentrapped pentacene was removed by centrifugation (13,000 rpm for 10 min) followed by a washing step at 3,000 rpm for 5 minutes. The final dispersion was subsequently filtered through a 0.2 μm pore membrane filter." In this passage, it is not clear where unentrapped pentacene was? Was it in the pellet?
Author’s response: we have modified the manuscript to better explain the unentrapped PTC separation process.
6. The materials and methods section lacks the whole part about albumin nanodots interaction, which should definitely be added. Especially, the excitation wavelength. It is not shown what kind of interaction do Cdots and BSA exhibit. The quenching of fluorescence can be both static and dynamic. The measurements should be repeated at three different temperatures to determine what kind of quenching this is. Maybe in this case it is a purely dynamic quenching. Potentially Cdots could possess unpaired electrons which could easily quench fluorescence. Also, authors should record spectra of just Cdots at the same excitation wavelength and in the same emission range to see if there would be some overlap of Cdots and BSA emission.
Author’s response: as suggested in the revised manuscript we included more experimental details about the albumin C-nanodot fluorescence measurements including excitation wavelength and emission spectra of C-nanodots at same excitation λ (280nm). Regarding the fluorescence at different temperature unfortunately at moment we cannot perform fluorescence measurement at various temperatures.
7. Ksv values should have units of ml/mg.
Author’s response: the typos was corrected
8. Figure 3 - It would be nice to highlight the moment when the irradiation stops. To show that the effect arises mostly from Cdots, a control experiment should be done with water only or even beta-CD dissolved in water.
Author’s response: thanks for your suggestion figure 3 was revised as suggested
9. In Supplementary material "tau s" is mentioned in the text, but does not appear in the equation 4. Is tau or tau s supposed to be shown in all of the following graphs? In the main text, "pi s" denotes the time constant. The use of variable names is inconsistent.
Author’s response: the typos was corrected
10. All the graphs in Figure S3 miss the letter theta in the x-axis title.
Author’s response: figures S3 were corrected as suggested
11. In Supplementary material it is not explained what hA parameter stands for. Lines 196-198 - "Additional emission bands at around 800-820 nm related to the PTC excitation (680 nm) were observed to again confirm the presence of the PTC in Cdots-βCD/PCT nanostrutcures" - A fluorescence spectrum at the 680 nm excitation should be shown for the Cdots without PTC as well.
Author’s response: as suggested the mission spectrum at excitation wavelength at 680 nm for Cdots-βCD and for Cdots-βCD/PTC were included in the revised manuscript.
12. Lines 211-213: "The PTC spectrum (blue curve) revealed these diagnostic peaks: 1700-1600 cm-1 (C=O stretching), 1570 cm-1 (aromatic C=O stretching), 1485 cm-1 (C=C stretching), 1350 cm-1 (C-N stretching), 1150-1100 cm-1 (C-Cl stretching)." Why would there be C-N and C-Cl stretching, if pentacene contains only carbons and hydrogens? The assignation should be redone and some more comments about FTIR spectra should be made.
Author’s response: the text was revised as suggested
13. Lines 214-219 - How is a 10-fold difference between DLS and AFM results possible, especially taking into account that the caption of Figure 4 states that the 3-D AFM image is a representative example.
Author’s response: thanks for your comment, probably the PTC molecules promote hydrophobic interactions between the Cdots-βCD/PTC nanostructures resulting in a net increase of hydrodynamic size in water. In the revised manuscript we have detailed this point.
14. Line 239 - Again time constant probably should not be "pi".
Author’s response: the typos was corrected
15. Line 253 - There is no Figure S11.
Author’s response: the typos was corrected
16. Why was the interaction between PTC Cdots and albumin not studied? Conclusions - Lines 396-399 - "As a support for potential biological applications, the low cytotoxicity of the Cdots-βCD/PCT nanohybrid was demonstrated as well as its ability to bind albumin with a strong affinity (association constant of about 2.78 ± 0.28398 mg ml-1), which suggests a potential role of the nanohybrid in the formation of corone proteins." - The strong affinity for BSA binding in the view of previous comments was not proved and Ksv is not an affinity constant.
Author’s response: Thanks for your suggestion a detailed investigation of the interaction between the Cdots-βCD/PTC and biological molecules as DNA, protein and cells will be reported in a further work. We revised the manuscript to better explain this point.
17. Lines 402-404 - "The results presented in this work, highlight the potential of Cdots-βCD/PTC as a novel versatile nanosystem for biomedical applications, such as visualization and site-specific photothermal treatment of cancer cells." - Since the colon cancer cell line was already used it would make sense to show using the same cell line the photothermal effect on cell viability.
Author’s response: a detailed application of Cdots-βCD/PTC as photothermal agent for cancer cell damage will be focused and reported in a further work.
Round 2
Reviewer 1 Report
Comments and Suggestions for Authors
This work can be accepted.
Author Response
Reviewer comment: This work can be accepted.
Reviewer 2 Report
Comments and Suggestions for Authors
The manuscript was somewhat improved after revision, however, some points have not been fully addressed, or at all. Thus, the manuscript should undergo either another major revision or be rejected.
The most important point from a previous review ("From the start of the manuscript it is not clear why Cdots are used as a carrier and not some kind of untreated cyclodextrin. What does one gain by converting cyclodextrins into Cdots and is this thermal synthesis procedure such that some (minor/major) portion of beta-CD remains intact, which consequently enables host-guest interactions?") was completely ignored.
"5. Lines 307-310: "The unentrapped pentacene was removed by centrifugation (13,000 rpm for 10 min) followed by a washing step at 3,000 rpm for 5 minutes. The final dispersion was subsequently filtered through a 0.2 μm pore membrane filter." In this passage, it is not clear where unentrapped pentacene was? Was it in the pellet?
________
Author’s response: we have modified the manuscript to better explain the unentrapped PTC separation process." REVIEWER COMMENT: The new text does not provide any new information and still, it is unclear what was done to purify the sample. If the precipitate after centrifugation should be preserved, and the unentrapped pentacene is in solution, which is the conclusion that can be made based on the fact that there was a subsequent washing step, how was it then filtered through a membrane? If unentrapped penatcene was, on the other hand in the precipitate, how would the supernatant containing cyclodextrin be washed?
_________
"In Supplementary material "tau s" is mentioned in the text, but does not appear in the equation 4. Is tau or tau s supposed to be shown in all of the following graphs? In the main text, "pi s" denotes the time constant. The use of variable names is inconsistent.
Author’s response: the typos was corrected"- REVIEWER COMMENT: The inconsistency in variable designation was not corrected in the supplementary material, also, the following remark was skipped: "In Supplementary material it is not explained what hA parameter stands for."
____________
In the revised manuscript there are no references for ATR-FTIR assignment in case of Cdots-betaCD-PTC, and they should be added to the text.
_____________
"13. Lines 214-219 - How is a 10-fold difference between DLS and AFM results possible, especially taking into account that the caption of Figure 4 states that the 3-D AFM image is a representative example.
Author’s response: thanks for your comment, probably the PTC molecules promote hydrophobic interactions between the Cdots-βCD/PTC nanostructures resulting in a net increase of hydrodynamic size in water. In the revised manuscript we have detailed this point. " - REVIEWER COMMENT: This point in the revised manuscript was not detailed other than extending the sentence with the statement that there are aggregates. Also, deleting the preexisting data raises suspicions.
Comments on the Quality of English Language
There are still some typos.
Author Response
The manuscript was somewhat improved after revision, however, some points have not been fully addressed, or at all. Thus, the manuscript should undergo either another major revision or be rejected.
The most important point from a previous review ("From the start of the manuscript it is not clear why Cdots are used as a carrier and not some kind of untreated cyclodextrin. What does one gain by converting cyclodextrins into Cdots and is this thermal synthesis procedure such that some (minor/major) portion of beta-CD remains intact, which consequently enables host-guest interactions?") was completely ignored.
Author ‘s response: In the manuscript, it was reported that Cdots-βCD possess luminescence and photothermal properties, which are known to be absent in untreated βCDs. Therefore, it follows that, unlike untreated βCDs, βCD-derived carbon dots offer the advantage of combining the luminescence and photothermal properties of carbon dots with the host properties of CDs for drug delivery. To this end, the procedure for the preparation of the carbon dots was optimized to provide a carbon core capped by a β-CD shell. The presence of intact βCDs on the shell was confirmed by NMR spectra of the Cdots-βCD that showed signals (e.g. H3 and H5 protons located within the cyclodextrin cavity) in the typical regions of the β-CD precursor. The presence of intact βCDs was corroborated by the ability to complex pentacene, a hydrophobic drug. The inclusion of pentacene in the βCD-derived carbon dots provided a novel photothermal nanosystem activable with biocompatible light (680 nm). This property makes the pentacene/carbon dots attractive for applications in anticancer photothermal therapy (PTT). Compared to untreated cyclodextrins, another advantage of CD-based carbon dots as carriers comes from their nanosize, which is known to face preferential accumulation in cancer tissues via EPR effect. In the revised manuscript, we added information on the advantage of Cdots-βCD over βCD and on the intact structure of βCDs in the shell.
"5. Lines 307-310: "The unentrapped pentacene was removed by centrifugation (13,000 rpm for 10 min) followed by a washing step at 3,000 rpm for 5 minutes. The final dispersion was subsequently filtered through a 0.2 μm pore membrane filter." In this passage, it is not clear where unentrapped pentacene was? Was it in the pellet?
Author ‘s response: The manuscript was modified to better explain the used procedure as follow: The solid unentrapped pentacene was removed by centrifugation (13,000 rpm for 10 min). The supernatant containing Cdots-βCD/PTC was separated by the pellet containing the unentrapped PTC and subsequently filtered through a 0.2 μm pore membrane.
Author’s response: we have modified the manuscript to better explain the unentrapped PTC separation process." REVIEWER COMMENT: The new text does not provide any new information and still, it is unclear what was done to purify the sample. If the precipitate after centrifugation should be preserved, and the unentrapped pentacene is in solution, which is the conclusion that can be made based on the fact that there was a subsequent washing step, how was it then filtered through a membrane? If unentrapped penatcene was, on the other hand in the precipitate, how would the supernatant containing cyclodextrin be washed?
Author ‘s response: The manuscript was modified to better explain the used procedure as follow: The solid unentrapped pentacene was removed by centrifugation (13,000 rpm for 10 min). The supernatant containing Cdots-βCD/PTC was separated by the pellet containing the unentrapped PTC and subsequently filtered through a 0.2 μm pore membrane.
_________
"In Supplementary material "tau s" is mentioned in the text, but does not appear in the equation 4. Is tau or tau s supposed to be shown in all of the following graphs?
Author ‘s response: The tau-s (τs) is now reported in both the manuscript and the equation 4 in SI
In the main text, "pi s" denotes the time constant. The use of variable names is inconsistent.
Author ‘s response: in the text is reported just tau-s (τs) no pi-s
Author’s response: the typos was corrected"- REVIEWER COMMENT: The inconsistency in variable designation was not corrected in the supplementary material, also, the following remark was skipped: "In Supplementary material it is not explained what hA parameter stands for."
Author response: h and A are the heat transfer coefficient and the surface area of the container respectively. Supporting Information section was properly modified as suggested.
____________
In the revised manuscript there are no references for ATR-FTIR assignment in case of Cdots-betaCD-PTC, and they should be added to the text.
Author response: This is the first example of Cdots-betaCD/PTC reported in literature, so no examples of FTIR spectrum are available. For this reason, in the original manuscript, we have reported and detailed the FTIR spectra for the precursors: PCT and CDs-βCD. Anyway, as suggested in the revised manuscript we have included a new reference for PTC and a reference for similar cyclodextrin derivate Cdots.
_____________
"13. Lines 214-219 - How is a 10-fold difference between DLS and AFM results possible, especially taking into account that the caption of Figure 4 states that the 3-D AFM image is a representative example.
Author’s response: thanks for your comment, probably the PTC molecules promote hydrophobic interactions between the Cdots-βCD/PTC nanostructures resulting in a net increase of hydrodynamic size in water. In the revised manuscript we have detailed this point. " - REVIEWER COMMENT: This point in the revised manuscript was not detailed other than extending the sentence with the statement that there are aggregates. Also, deleting the preexisting data raises suspicions
Author response: The data was not deleted; to demonstrate, without suspicions, the presence of large aggregates in the last revision we had included the DLS distribution graphs for both Cdots-βCD and Cdots-βCD/PTC. Moreover, the original text reports the presence of large aggregates. To investigate the mode and the strength of the interaction between PTC species is out of the scope of this work. We have modified the sentence to better explained difference between DLS and AFM measurement.
Round 3
Reviewer 2 Report
Comments and Suggestions for Authors
The authors probably misunderstood that in the previous review, their responses to the first review were quoted along with the first reviewer's comments (even though the text was between the quotation marks). This is just to highlight that the reviewer carefully reviewed all of the corrections made to the manuscript and supplementary information. After the latest corrections manuscript is significantly improved, however, it still needs English and typo corrections. Also, the following two minor points need to be addressed, after which the manuscript can be published.
Lines 220-223 and 360-363: The supernatant containing Cdots-βCD/PTC was separated by the pellet containing the unentrapped PTC and subsequently filtered through a 0.2 μm pore membrane. - It should be then "separated from" instead of "separated by".
AUTHORS: Author response: h and A are the heat transfer coefficient and the surface area of the container respectively. Supporting Information section was properly modified as suggested.
REVIEWER: Maybe the corrected version was not uploaded properly, since in the latest supporting information there is still no explanation of the hA parameter meaning.
Comments on the Quality of English LanguageFixing of typos
Author Response
Dear Editor,
please find enclosed the revised manuscript entitled “Emissive Pentacene-Loaded βcyclodextrin-derived C-nanodots Exhibit Red-Light Triggered Photothermal effect.” by Ludovica Maugeri, Giorgia Fangano, Ester Butera, Giuseppe Forte, Paolo Giuseppe Bonacci, Nicolo’ Musso, Francesco Ruffino, Loredana Ferreri, Grazia Maria Letizia Consoli and myself. The modification in the text have been highlighted in red fonts.